# Timing of ICSI with Respect to Meiotic Spindle Status

**DOI:** 10.3390/ijms24010105

**Published:** 2022-12-21

**Authors:** Olga Tepla, Zinovij Topurko, Simona Jirsova, Martina Moosova, Eva Fajmonova, Radek Cabela, Katerina Komrskova, Irena Kratochvilova, Jaromir Masata

**Affiliations:** 1Department of Obstetrics and Gynaecology First Faculty of Medicine, Charles University and General University Hospital in Prague, Apolinarska 18, 128 08 Prague, Czech Republic; 2Laboratory of Reproductive Biology, Institute of Biotechnology of the Czech Academy of Sciences, BIOCEV, Prumyslova 595, 252 50 Vestec, Czech Republic; 3Department of Zoology, Faculty of Science, Charles University, Vinicna 7, 128 44 Prague, Czech Republic; 4Institute of Physics of the Czech Academy of Sciences, Na Slovance 2, 182 21 Prague, Czech Republic

**Keywords:** human oocyte, meiotic spindle, polar body, in vitro fertilization, ICSI, gravidity, polarized light microscopy

## Abstract

The aim of this study was to evaluate the efficiency of using meiotic spindle (MS) visibility and relative position to the polar body (PB) as indicators of oocyte maturation in order to optimize intracytoplasmic sperm injection (ICSI) timing. This was a cohort study of patients younger than 40 years with planned ICSI, the timing of which was determined by MS status, compared with those without MS evaluation. The angle between PB and MS and MS visibility were evaluated by optical microscope with polarizing filter. Oocytes with MS evaluation were fertilized according to MS status either 5–6 h after ovum pick-up (OPU) or 7–8 h after OPU. Oocytes without MS evaluation were all fertilized 5–6 h after OPU. For patients over 35 years visualization of MS influenced pregnancy rate (PR): 182 patients with MS visualization had 32% PR (58/182); while 195 patients without MS visualization had 24% PR (47/195). For patients under 35 years, visualization of MS did not influence PR: 140 patients with MS visualization had 41% PR (58/140), while 162 patients without MS visualization had 41% PR (66/162). Visualization of MS therefore appears to be a useful parameter for assessment of oocyte maturity and ICSI timing for patients older than 35.

## 1. Introduction

In the in vitro fertilization (IVF) process, the quality of the human oocytes is a crucial parameter [1,2,3,4,5]. IVF success rate (with up to 30% pregnancy rate in one cycle) is affected by several key factors [3,6,7]: the age of the women, oocyte quality, sperm quality, and the full maturation of the oocytes [8]. The ability to improve the non-invasive selection of developmentally mature human oocytes may increase the overall efficiency of human assisted reproduction.

The presence of polar body (PB) is generally considered to be a marker of oocyte nuclear maturity and preparation for fertilization. However, recent studies have shown that oocytes displaying a PB may still be immature (at the early Telophase I stage) [2,9,10] even though PB is present in the perivitelline space. In this stage, there is continuity between the ooplasm of the oocyte and the formation of PB and the MS is interposed between two separating cells [3,6,11,12,13,14,15]. Physiologically this step has a duration of 75–90 min. MS have been found to disappear in late Telophase I but can reform 40–60 min later [16] at the stage of the oocyte interphase between the first and second meiotic division [17,18]. L. Rienzi et al. [2,10,13,14] studied the relationship between MS/PB position and oocyte potential after ICSI. In Telophase I, the first polar body can be visible, but cytokinesis is not yet finished, and the cytoplasmic bridge between the oocyte and polar body still exists. In this cytoplasmic bridge, the spindle with the homologous chromosomes can be detected. Meiosis II initiates immediately after cytokinesis, usually before the chromosomes have fully decondensed; the first MS disappears and is not visible for about 1–1.5 h. The second MS reappears at metaphase II, and the chromosomes align on the spindle with microtubules from opposite poles of the spindle attached to the kinetochores of sister chromatids. MS is a microtubular structure involved in chromosome segregation, and, therefore, it is crucial in the sequence of events leading to the correct completion of meiosis and subsequent fertilization. Parallel-aligned MS microtubules are birefringent and are able to shift plane polarized light inducing a retardance [19,20,21,22]; birefringent MS properties enable the polarized light microscope to generate contrast and image MS structure [23]. Therefore, MS status provides the embryologist with more accurate information about the nuclear stage of the oocyte [20,24,25,26].

Several authors [24,25,26,27,28] suggest that the presence, position, and retardance of the optically birefringent MS are related to developmental oocyte competence. Some studies [19,20,21,22] have proposed using polarized light microscopy to adjust the timing of intracytoplasmic sperm injection (ICSI) based on MS morphology, on the principle that the strategy can be beneficial for late-maturing oocytes. Oocyte meiotic spindle morphology has also been evaluated as a predictor of blastocyst ploidy [29]. A strong association of MS morphology with fertilization, cleavage to at least six cells on day 3, and good/top-quality blastocyst formation was presented.

The aim of our study was to evaluate the efficiency of a method to assess the optimal time for oocyte fertilization by ICSI. Finding the right time for ICSI by using different parameters has been studied recently [5,30,31,32,33,34,35]. The essence of the method is the use of the position of oocytes OP and MS together with MS visibility as indicators of oocyte maturity and consequently as pointers for proper ICSI timing. The angle between PB and MS together with MS visibility were non-invasively evaluated by optical microscope with polarizing filter.

## 2. Results

### 2.1. MS Used as an Indicator of Oocyte Maturation

Of 679 IVF patients (average age 34.7 years, average number of oocytes in one OPU 4.2), 377 patients were older than 35 years and younger than 40 years (average age 36.9 years, average number of oocytes in one OPU 3.9) and 302 patients were younger than 35 years (average age 31.9 years, average number of oocytes in one OPU 4.6), see Table 1. We decided to divide patients into two age dependent groups: 1. Group of patients over 35 years (≥35 years). 2. Group of patients younger than 35 years (<35 years). The reason for this division was a significant change in pregnancy rate around patient age of 35 (Appendix A, Table 1). The 377 patients older than 35 (1418 collected oocytes) had a pregnancy rate of 28% (105/377), and their average utilization rate was 53%. The 302 patients younger than 35 (1211 collected oocytes) had a pregnancy rate of 41% (124/302), and their average utilization rate was 45%. The differences in pregnancy rates between patients < 40 treated with MS evaluation (116/32) and patients <40 treated without MS evaluation (113/357) was not statistically significant: see Table 1.

#### 2.1.1. MS Used as an Indicator of Oocyte Maturation for Patients Younger Than 35 Years

The 140 patients younger than 35 whose 681 oocytes were evaluated by MS visualization (average age 32.1 years, average number of oocytes in one OPU 4.4) with the ICSI time modified according to MS status became pregnant in 41% of cases (58/140), and their average utilization rate was 49% (Figure 1 and Figure 2). The 162 patients younger than 35 treated without MS visualization (average age 31.7 years, average number of oocytes in one OPU 4.7) became pregnant in 41% of cases (66/162), and their average utilization rate was 45%. The differences between pregnancy and average utilization rates for patients < 35 treated with MS evaluation and without MS evaluation was not statistically relevant (*p* > 0.1) [36].

The differences in pregnancy rates between patients < 35 treated with MS evaluation and with ICSI performed 5–6 h after OPU (pregnancy rate 43%) and patients < 35 treated with MS evaluation and with ICSI performed 7–8 h after OPU (pregnancy rate 40%) was not statistically relevant (*p* > 0.1), see Table 2. From these results it is clear that the evaluation of MS using an optical microscope with a polarizing filter does not increase stress on the oocytes and does not negatively affect pregnancy and average utilization rates.

#### 2.1.2. MS Used as an Indicator of Oocyte Maturation for Patients Older Than 35 Years

A different situation is found for the patients ≥ 35 years (Figure 1). A total of 32% (58/182) of patients treated with MS evaluation (ICSI time modified according to MS status) became pregnant, and their average oocyte utilization rate was 53% (699 oocytes, average age 36.2 years, average number of oocytes in one OPU 4.1). Just 24% of patients ≥35 treated without MS evaluation (719 oocytes, average age 37.2 years, average number of oocytes in one OPU 3.7) became pregnant (47/195); the average utilization rate was 42% (Table 1 and Appendix A).

The value of the z-test (the difference between pregnancy rates of patients with and without MS evaluation) was −1.73. The value of *p* was 0.08. For the differences in average utilization rates between groups with and without MS evaluation the *p* value was <0.05. (The value of z was −2.1, the value of *p* was 0.03).

For the differences in pregnancy rates between MS evaluated patients ≥ 35 with ICSI proceeded 5–6 h after OPU (pregnancy rate 37%—Table 2) and MS evaluated patients ≥ 35 with ICSI performed 7–8 h after OPU (pregnancy rate 27%—Table 2) the *p* value was smaller than 0.1. MS evaluated patients with ICSI performed 5–6 h after OPU had a higher relative number of oocytes with MS clearly visible and PB/MS not in close proximity (>40%) compared to MS evaluated patients with ICSI performed 7–8 h after OPU. It should be noted that the clear visibility and not close proximity of PB/MS can indicate high quality of oocytes correlating to a high (37%) pregnancy rate. Even for MS evaluated patients ≥ 35 with later fertilization (7–8 h after OPU, >60% oocytes with MS not visible or with PB/MS in close proximity) the pregnancy rate was higher (27%, see Table 2) than pregnancy rate of all patients ≥ 35 and <40 treated without MS evaluation (24% pregnancy rate, see Table 1).

For the differences in pregnancy rates between MS evaluated patients < 35 with ICSI performed 5–6 h after OPU (pregnancy rate 43%, see Table 2) and MS evaluated patients < 35 with ICSI performed 7–8 h after OPU (pregnancy rate 40%—Table 2) the *p* value was higher than 0.1.

Even though the statistical relevance (*p*) was <0.1 in the case of the differences in pregnancy rates between MS evaluated patients ≥ 35 with ICSI performed 5–6 h after OPU and MS evaluated patients ≥ 35 with ICSI performed 7–8 h after OPU (number of patients in the individual groups), we regard the results as clinically significant. It is important to continue this research with a bigger cohort and with the help of artificial intelligence and neural network learning to achieve advanced PB/MS evaluation.

To complete the statistics, we also compared pregnancy and utilization rates of patients treated with MS evaluation and patients treated without MS evaluation (Figure 2). In the case of 322 patients (1380 oocytes) whose oocytes were evaluated by MS visualization, the pregnancy rate was 36% (116/322). Of 357 patients, whose 1249 oocytes were not evaluated by MS visualization, 113 became pregnant (32%). Our data correlated with data published in [3,6,7]. The difference of pregnancy rates between patients treated with MS visualization and those without MS visualization was not significant: the value of z was 1.01 and the value of *p* was 0.27. For further results and correlations, see Appendix A.

### 2.2. Correlations between Relevant Parameters

There was no relevant correlation between sperm quality and utilization rates and between sperm quality and pregnancy rates for all the groups and patients in the study. For all the patients the value of the correlation coefficient between sperm quality and pregnancy rates was −0.14, *p* > 0.05 and the value of the correlation coefficient between sperm quality and utilization rate was −0.12, *p* > 0.05. The *t*-value representing the significance of the difference between sperm quality of partners of patients treated with MS evaluation and sperm quality of partners of patients treated without MS evaluation was 0.45, and the *p*-value was 0.33.

There was no relevant correlation between sperm quality and utilization rates and between sperm quality and pregnancy rates for all the groups and patients under study. For all the patients the value of the correlation coefficient between sperm quality and pregnancy rates was −0.14, *p* > 0.05 and value of the correlation coefficient between sperm quality and utilization rate was −0.12, *p* > 0.05. The *t*-value representing the significance of the difference between sperm quality of partners of patients treated with MS evaluation and sperm quality of partners of patients treated without MS evaluation was 0.45, and the *p*-value was 0.33.

The *t*-value representing the significance of the difference between the age of patients treated with MS evaluation and age of patients treated without MS evaluation was −0.5, the *p*-value was 0.3. Correlation matrixes calculated for ICSI timing, sperm quality, utilization rates, pregnancy rates, number of GV, and M1 for all patients are presented in Appendix A. For each age group, there was no relevant correlation between sperm quality, number of GV, and M1 and utilization/pregnancy rates. The strongest correlation was between ICSI timing and pregnancy rates of patients ≥ 35 years (Appendix A) confirming the impact of MS evaluation and consequent time tuning for patients ≥35 years.

## 3. Discussion

In this work the MS status (visibility and relative position to PB) was used as an indicator of oocyte maturation and, consequently, as an indicator of optimal ICSI timing with respect to patient’s age. The angle between PB and MS together with MS visibility were non-invasively evaluated by optical microscope with polarizing filter.

We compared utilization and pregnancy rates of patients younger than 40 years and treated with MS evaluation and without MS evaluation. On the basis of MS status ICSI was performed 5–6 h after OPU or 7–8 h after OPU. Oocytes without MS evaluation were in all cases fertilized 5–6 h after OPU. According to our results, the age of patients was an important parameter that determined the impact of MS state on the utilization and pregnancy rates. Postponing the ICSI time by 2–3 h (7–8 h after OPU) for the purpose of sufficient maturation [21] of oocytes, was based on the strategy proposed in [2,13,15,20,28]. That is, timing 7–8 h after OPU is based on [21] according to which ICSI should be performed on the day of OPU and should not exceed 9 h.

The difference in pregnancy rate of patients ≥ 35 treated with MS evaluation and treated without MS evaluation was clinically relevant: 32% patients ≥ 35 treated with MS evaluation became pregnant and just 24% of patients ≥ 35 treated without MS evaluation became pregnant. For patients older than 35 years the oocytes maturation was typically less naturally synchronized and modifying ICSI time according to MS status as an indicator of oocytes maturation significantly increased the overall efficiency of human assisted reproduction [6,7,12,37,38]. For the oocytes of patients ≥ 35 years, the longer incubation time enabled a greater number of oocytes to become fully mature (MII phase) and prepared for ICSI. In this situation, the MS status is an important indicator of oocyte immaturity. In contrast, for patients younger than 35 years the oocyte maturation was naturally synchronized [6,7,39] and the modifying ICSI time according to MS status did not significantly influence the efficiency of human assisted reproduction. However, the fact that for patients < 35 modifying ICSI time according to MS status did not significantly influence the efficiency of human assisted reproduction supports the claim that the evaluation of MS does not increase stress on the oocytes and does not negatively affect pregnancy and utilization rates.

PB-displaying oocytes might be engaged in chromosome segregation or early phases of spindle reconstitution (interkinesis) and are thus not ready for fertilization. The presumption was that in the relevant part of the oocytes the absence of spindle birefringence or close position of PB and MS might be only temporary and can be a sign of immaturity. Indeed, more than half of spindle-missing oocytes managed to form a detectable MII spindle when ICSI was postponed to a later time. The delay in spindle formation can be explained by the fact that, in the absence of centrosome, microtubule nucleation is slow, and it may take a couple of hours for a bipolar MII spindle to be reformed after PB extrusion [1,2]. Elderly patients may experience a slowdown in maturation. Performing ICSI on an oocyte that has not reached MII often leads to failure of fertilization, asynchrony of the biochemical processes of fertilization, an irregular number of pronuclei, or the arrest of embryonic development before the blastocyst stage. On the other hand, postponing the time of ICSI by 2–3 h according to MS status did not appear to have a negative effect on the pregnancy and utilization rates.

## 4. Materials and Methods

### 4.1. Patients and Collection of Oocytes

Women under the age of 40, who underwent ICSI from January 2021 to March 2022 in the Centre of Assisted Reproduction (CAR), Department of Obstetrics and Gynaecology First Faculty of Medicine, Charles University and General University Hospital in Prague, Prague 2, Czech Republic, were consecutively included in the study. The study was conducted in accordance with the Declaration of Helsinki. All the participating subjects included in the cohort study accepted procedures used by the Department of Obstetrics and Gynaecology First Faculty of Medicine, Charles University and General University Hospital in Prague. This study was approved by the Ethics Committee of the General University Hospital, Prague (1821/20 S-IV).

To induce oocyte maturation, we used the application of 250 ug hCG, i.e., approx. 6500 IU (Ovitrelle Merck Serono) or 0.2 mg of GnRH agonist triptoreline (Diphereline Ipsen Pharma). After the trigger, the follicular fluid was punctured between 36 and 37 h according to the standard protocol.

The key criteria to be included in the study were ICSI fertilization and a spermiogram containing at least 0.5 mil/mL of progressively motile spermatozoa at the time of ICSI. All patients involved in the study (undergoing ICSI) were divided into two age dependent groups: 1. patients older than 35 years (after their 35th birthdays) and younger than 40 years, 2. patients younger than 35 years.

In the group of patients in which the MS was determined by polarized microscopy the position of PB and MS together with MS visibility were used as indicators of oocyte maturity. It should be noted that patients with MS evaluated by polarized microscopy were randomly selected, and only the patients who signed the agreement (more than 90% of patients) with the spindle status observation formed the MS evaluated group. The control group was formed by the standard patients undergoing ICSI. Female patients over 40 years of age were not included in the research due to the fact that at this age the scale of factors negatively affecting the pregnancy rate is very high, i.e., fertility in this age category is very low.

Using an optical microscope with a Nikon CEE GmbH polarizing filter, the angle between PB and MS together with MS visibility were determined. The photos from microscope of Nikon Europe B.V., Czech Branch Office, Microscope Nikon Eclipse Ti2 Series, Nikon Spindle Observation System are presented as illustration of the methodology (Figure 3). Further, the illustrative using microscope with a Nikon CEE GmbH polarizing filter in the Centre of Assisted Reproduction (CAR), Department of Obstetrics and Gynaecology First Faculty of Medicine, Charles University and General University Hospital in Prague, Prague 2, Czech Republic (Appendix A). Oocyte assessment was performed by two embryologists to ensure objective evaluation. The oocytes were rotated with the needle during observation so that the angle between the polar body and the MS was well evaluated. All the patients with determined oocytes meiotic spindles status were divided into two groups: 1. Patients having more than 60% oocytes with PB/MS in close proximity (angle between PB and MS < 5°) (Figure 3A, Appendix A) or MS not visible (Figure 3B and Appendix A) were predicted as immature oocytes. For these patients ICSI was performed 4–5 h after the polarization microscopy evaluation, i.e., 7–8 h after OPU. 2. Patients having at least 40% oocytes with MS clearly visible and PB/MS not in close proximity (angle between PB and MS > 5°) (Figure 3C and Appendix A) were predicted as mature oocytes (Figure 3C, Appendix A). For these patients ICSI was performed typically 2–3 h after the polarization microscopy evaluation, i.e., 5–6 h after OPU, as patients from control group. The reasons for this process strategy are operational: in our center it is required that all oocytes of one patient be fertilized at the same time/in one group.

Postponing the ICSI time by 2–3 h (7–8 h after OPU) for the purpose of sufficient maturation [21] of oocytes, was based on the strategy proposed in [2,13,15,20,28]. That is, timing 7–8 h after OPU was based on [21] according to which ICSI should be performed on the day of OPU and should not exceed 9 h. Oocytes without MS evaluation were in all cases fertilized 5–6 h after OPU.

The probability of obtaining high-quality embryos (utilization rate) and clinical pregnancy (pregnancy rate) for patients with the time of ICSI determined on the basis of the relative position of PB and MS and the MS visibility were compared with the probability of obtaining high-quality embryos and clinical pregnancy for patients without MS evaluation. Since the average number of oocytes collected from one patient in one OPU was 4–5, we used 40% of the oocytes collected from one patient with PB/MS not in close proximity and MS clear visibility as a limit for the decision whether ICSI was performed 5–6 h after OPU or patients ICSI was performed 7–8 h after OPU. Typically, the angle between PB and MS began to increase approximately after two hours for oocytes with the MS and PB in close proximity (Figure 1, Figure 2 and Appendix A).

### 4.2. Sperm Quality

The quality of sperm was monitored so that correlation between sperm quality and utilization and pregnancy rates could be evaluated: (N) normozoospermia (46%); (A) asthenozoospermia (10%); (O) oligospermia (10%); (OA) oligoastenozoospermia (4%); (OTA) Oligo-astheno-teratozoospermia (30%) [30].

### 4.3. Oocytes Treatment

ICSI was performed according to standard protocol using ICSI/holding micropipettes (#00-30/#001-120-30, Microtech IVF, Czech Republic), polyvinylpyrrolidone (ICSI™, Vitrolife, Sweden), and Eppendorf (Hamburg, Germany) micromanipulation system equipped with thermoplate (Tokaohit, Japan).

The oocytes were denuded (HYASE-10X™, Vitrolife, Sweden) after OPU, and the maturation stage was examined. GV stage oocyte were not included. Oocytes were cultivated in a 4-well plate (NUNC) using Sage Fertilization Medium™, Origio, Denmark, Sage 1-Step™, Origio, Denmark, under paraffin oil (OVOIL™, VITROLIFE, Sweden) at 37.0 °C and 6% CO_2_.

### 4.4. Embryo Cultivation

Fertilization after ICSI was defined as the presence of two pronuclei and 2 polar body 16–20 h post ICSI. Embryos were cultivated for 2–6 days. For transfer and freezing we used good quality embryos. The probability of obtaining high-quality embryos (utilization rate) and clinical pregnancy after first OPU was compared between these two subgroups [30]. We freeze embryos at our clinic on day 5 and day 6 (in 98% of cases). Patients whose embryos were of sufficient quality for transfer were included in the study: cleavage embryos—gr.1–2 [31] and blastocyst with good expansion of blastocele cavity and the integrity of both the inner cell mass and trophectoderm cells [32,33,34]. Embryos were cultivated in a 4-well plate (NUNC) using Sage 1-Step™, Origio [33,34,35], Denmark, under paraffin oil (OVOIL™, Vitrolife, Sweden) at 37 °C and 6% CO_2_.

Out of the total number of evaluated and transferred embryos, there were 34% of day 4 embryos, 14% of day 5 embryos, and 17% of day 3 embryos. The transfer time was modified according to the condition of the patient and the state of embryo. Predominantly only 1 embryo was used for the embryo transfer, and the average of transferred embryos per one cycle was 1.03.

### 4.5. IVF Outputs under Consideration

During the presented research the following IVF outputs for all patients in the study were recorded: (1) Oocyte utilization rates—the number of high-quality embryos (for transfer of freezing) divided by the number of all oocytes with a detected PB (fertilized by ICSI) calculated for each patient. Typically, it was from the part of oocytes with a detected PB (fertilized by ICSI) that high quality embryos were developed. Average utilization rates were calculated over each group of patients. (2) The patient clinical pregnancy rate was the number of pregnant patients from each group divided by the number of all patients in this group. A clinical pregnancy was confirmed by ultrasound visualization of the gestational sac or heartbeat. In our study the signs of clinical pregnancy were detected four weeks after embryo transfer.

### 4.6. Statistical Analysis

To quantify the statistical significance of the collected data, we calculated the *p* values as the rates of obtaining test results that were at least as extreme as the observed results during the test, assuming that the null hypothesis was correct (i.e., the rate that we falsely rejected the null hypothesis). The results were controlled by using the MaxStat Pro 3.6 program and Pearson Calculator [36]. Patient average utilization rates were calculated as an average for ratios of the number of quality embryos and the number of oocytes collected from each patient.

The t- and z- values as the significance of the difference between the compared groups of patients was calculated using the Pearson Calculator. The z-score and t-score show how many standard deviations there are from the mean of the distribution. The *t*-test is a statistical test that compares the mean and standard deviation of two samples to see if there is a significant difference between them. The z-score is a numerical measurement that describes a value’s relationship to the mean of a group of values. The t-values (normal distribution) and z-value (binary distribution) were calculated for all the compared group of patients according to their pregnancy rates and utilization rates. Correlation between sperm quality and both utilization rate and patient’s clinical pregnancy were also calculated using Pearson Correlation Coefficient Calculator [36].

## 5. Conclusions

The present study proved that visualization of MS is a useful parameter for assessment of oocyte maturity and ICSI timing for patients older than 35 years (based on MS visualization PR was 32%, compared with 24% without visualization). In contrast, the difference between pregnancy and average utilization rates for patients under 35 years treated with MS evaluation and without MS evaluation was not significant. We believe that the clinically significant results presented here will be valuable in implementing a follow-up study with image processing by artificial intelligence to implement PB/MS evaluation into clinical practice.

## Figures and Tables

**Figure 1 ijms-24-00105-f001:**
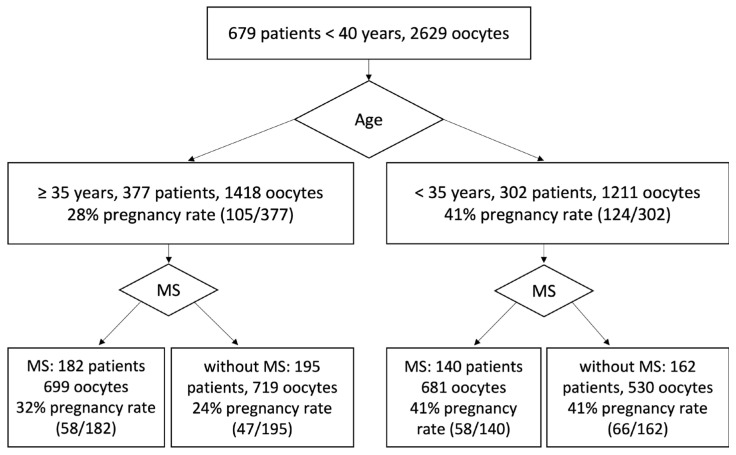
A total of 679 patients with planned ICSI fertilization were divided into two age dependent groups: 1. 377 patients older than 35 and younger than 40; 2. 302 patients younger than 35. For each age dependent group part of the patients had the time of ICSI determined on the basis of the relative position of PB and MS and the MS visibility. We compared the probability of clinical pregnancy for patients with the time of ICSI determined on the basis of the relative position of PB and MS and the MS visibility with the probability of clinical pregnancy for patients without MS evaluation.

**Figure 2 ijms-24-00105-f002:**
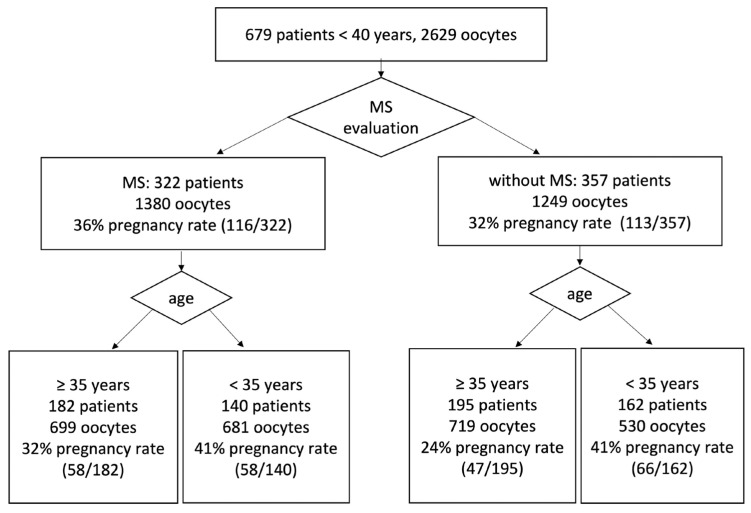
A total of 679 patients with planned ICSI fertilization were divided into two groups. In one group patients had time of ICSI determined on the basis of the relative position of PB and MS visibility. The second group was treated without MS evaluation. We compared the probability of clinical pregnancy for patients with the time of ICSI determined on the basis of the relative position of PB and MS and MS visibility with the probability of clinical pregnancy for patients without MS evaluation.

**Figure 3 ijms-24-00105-f003:**
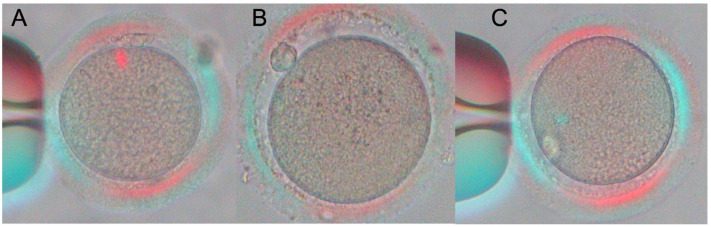
Oocyte maturation indicated by PB/MS proximity and MS visibility, captured by polarized light microscopy and presented without postprocessing adjustments. (**A**) The oocyte with PB and MS in close proximity; (**B**) the oocyte with no visible MS. (**C**) The oocyte with visible PB and MS not in close proximity, oocyte was rotated to determine the optimal morphology and location of MS. Photos were taken on microscope of Nikon Europe B.V., Czech Branch Office, Microscope Nikon Eclipse Ti2 Series, Nikon Spindle Observation System. Microscopic images were taken at ×100 magnification.

**Table 1 ijms-24-00105-t001:** Number of patients, their average ages, number of oocytes in group, average number of oocytes per OPU, and pregnancy rates calculated for groups of patients of different age and different treatment: treated either with MS evaluation or without MS evaluation. The significance values of the difference between pregnancy and utilization rates were calculated.

Group	Age (Years) Treatment	Pregnancy Rate	Number of Oocytes	Number of Patients	Related Group	*p* Value	Average Age (Years)
1	<35 years MS evaluation	41%	681	140	2	>0.1	32.1
2	<35 years without MS evaluation	41%	530	162	1	>0.1	31.7
3	≥35 and <40 years MS evaluation	32%	699	182	4	<0.1	36.2
4	≥35 and <40 years without MS evaluation	24%	719	195	3	<0.1	37.2
5	<40 years MS evaluation	36%	1380	322	6	>0.1	34.4
6	<40 years without MS evaluation	32%	1249	357	5	>0.1	34.7

**Table 2 ijms-24-00105-t002:** Number of pregnant patients, number of their oocytes, and pregnancy rates calculated for groups patients of different age and treated with MS evaluation. The significance values of the difference between pregnancy rates were calculated.

Group	Age (Years) Treatment	Pregnancy Rate	Number of Oocytes	Number of Patients	Related Group	*p* Value
1	≥35 and <40 years ICSI 5–6 h after OPU	37%	370	86	2	<0.1
2	≥35 and <40 years ICSI 7 h after OPU	27%	329	96	1	<0.1
3	<35 years ICSI 5–6 h after OPU	43%	331	67	4	>0.1
4	<35 years ICSI 7–8 h after OPU	40%	350	73	3	>0.1

## Data Availability

Data used to generate results in the paper are available in Appendix A.

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
