# Peer review of "Timing of ICSI with Respect to Meiotic Spindle Status"

_ijms, 2022, doi:10.3390/ijms24010105_

Round 1
Reviewer 1 Report
In this manuscript, the authors used meiotic spindle (MS) visibility and relative position to the polar body (PB) as indicators of oocyte maturation for intracytoplasmic sperm injection (ICSI) timing with respect to patient’s age. As the authors note, visualization of MS is useful parameter for assessment of oocyte maturity and ICSI timing for patients older than 35. This is a valuable project that will have the opportunity to be applied for the clinic. It could be published. Details need to be addressed:
Q1: L37 The reference is not quoted correctly, such as ref. 5. Please check whether the same error occurs in other locations.
Q2: L69-71 The abbreviations of the words were repeated twice without careful checking (ICSI).
Q3: Was there a method in previous studies to “evaluate the optimal oocyte fertilization time by ICSI based on patient age”?
Q4: Figure S2 in the supplementary material is incorrectly marked.
Q5: Is there any error in the data of 5 and 6 in Table 1? Please check it carefully
Q6: Is there already death in MS oocytes that are not visible? Are further tests conducted? (Fig.3)
Q7: L251-253 In the case of patients who did not sign the agreement with the spindle status observation, the status of the meiotic spindle was not determined. Could the authors discuss that the non-signers make a big difference?
Author Response
RESPONSE TO REVIEWERS
We thank all reviewers very much for evaluating our manuscript and providing us with useful notes about how to improve the article. We completely agree with all the recommendations and have therefore modified the manuscript accordingly. Step-by-step responses are provided below. All the relevant changes are highlighted in yellow.
Q1: L37 The reference is not quoted correctly, such as ref. 5. Please check whether the same error occurs in other locations.
Corrected.
Q2: L69-71 The abbreviations of the words were repeated twice without careful checking (ICSI).
Corrected.
Q3: Was there a method in previous studies to “evaluate the optimal oocyte fertilization time by ICSI based on patient age”?
Corrected: Finding the right time for ICSI by using different parameters has been studied recently [5].
In our work the MS status (visibility and a relative position to PB) was used as an indicator of oocyte maturation and, consequently, as an indicator of optimal ICSI timing with respect to patient’s age. Our presumption was that in the relevant part of the oocytes the absence of spindle birefringence or close position of PB and MS might be only temporary and can be a sign of immaturity. Indeed, more than half of spindle-missing oocytes managed to form a detectable MII spindle when ICSI was postponed to a later time. The delay in spindle formation can be explained by the fact that, in the absence of centrosome, microtubule nucleation is slow, and it may take a couple of hours for a bipolar MII spindle to be reformed after PB extrusion [1, 2]. Elderly patients may experience a slowdown in maturation. Performing ICSI on an oocyte that has not reached MII often leads to failure of fertilization, asynchrony of the biochemical processes of fertilization, an irregular number of pronuclei, or the arrest of embryonic development before the blastocyst stage. On the other hand, postponing the time of ICSI by 2-3 hours according to MS status did not appear to have a negative effect on the pregnancy and utilization rates.
Q4: Figure S2 in the supplementary material is incorrectly marked.
Corrected.
Q5: Is there any error in the data of 5 and 6 in Table 1? Please check it carefully
Checked. The differences in pregnancy rates between patients < 40 treated with MS evaluation (116/32) and patients < 40 treated without MS evaluation (113/357) was not statistically significant (p > 0.1), see Table 1.
Q6: Is there already death in MS oocytes that are not visible? Are further tests conducted? (Fig.3)
For the differences in pregnancy rates between MS evaluated patients ≥ 35 with ICSI proceeded 5-6 hours after OPU (pregnancy rate 37% - Table 2) and MS evaluated patients ≥ 35 with ICSI performed 7-8 hours after OPU (pregnancy rate 27% - Table 2) the p value was smaller than 0.1. MS evaluated patients with ICSI performed 5-6 hours after OPU had a higher relative number of oocytes with MS clearly visible and PB/MS not in close proximity (>40%) compared to MS evaluated patients with ICSI performed 7-8 hours after OPU. It should be noted that the clear visibility and not close proximity of PB/MS can indicate high quality of oocytes correlating to high (37%) pregnancy rate. Even for MS evaluated patients ≥ 35 with later fertilization (7-8 hours after OPU, >60% oocytes with MS not visible or with PB/MS in close proximity) the pregnancy rate was higher (27 %, see Tab. 2) than pregnancy rate of all patients ≥ 35 and <40 treated without MS evaluation (24% pregnancy rate, see Tab. 1).
Q7: L251-253 In the case of patients who did not sign the agreement with the spindle status observation, the status of the meiotic spindle was not determined. Could the authors discuss that the non-signers make a big difference?
In the group of patients in which the MS was determined by polarized microscopy the position of PB and MS together with MS visibility were used as indicators of oocyte maturity. It should be noted that patients with MS evaluated by polarized microscopy were randomly selected, and only the patients who signed the agreement (more than 90% of patients) with the spindle status observation formed the MS evaluated group. The control group was formed by the standard patients undergoing ICSI.
Best regards,
Prof. Irena Kratochvilova, Ph.D. and A/Prof. Katerina Komrskova, Ph.D.

Reviewer 2 Report
These researchers measured first whether a Meiotic spindle (MS) was visible and secondly the angle between the MS and the polar body (PB). They claimed that seeing the MS and having a wide angle between the MS and PB was a measure of oocyte maturity. and with certain cohorts of oocytes they delayed the timing of ICSI to allow for ‘further maturation’. Higher pregnancy rates were observed when ICSI was delayed (7-8h post OPU cf 5-6h) in cohorts of eggs from patients (>35yo) whereby “60% of oocytes with the PB/MS in close proximity or the MS was not visible” . The authors claimed that was due to maturity, but there is no proof of this.
Interesting data, but should be better presented if publication is to be considered.
Some major design faults
1. Only those consenting to MS got MS. Those not consenting were controls. This would create bias. All those consenting should have been randomised between having MS and not having MS
2. If ‘measuring maturity’ was an end point, then each oocyte should have been treated individually. Ie all oocytes with ‘the PB/MS in close proximity or the MS was not visible’ should have been split into early and late ICSIs, and paired within cohorts, and equally all oocytes with the ‘MS clearly visible and the PB and MS not in close proximity’ should have been split into the 2 ICSI times rather than the whole cohort (ie at least 40% oocytes with MS clearly visible, PB/MS not in close proximity).
3. All oocytes should have been followed up with fert data (1PN, 2PN, 3PN, 0PN, syngamy)-this would give a much clearer picture of maturity and timing of actual fertilisation
4. The stats were not clear. I have never seen P<0.1 being classed as significant,
5. Apart from age and sperm quality, there were no other demographic comparisons of the patient groups (stim type, # previous cycles, type of infertility, # oocytes, #GVs and M1s within a cohort, day of transfer etc) Note: this would not be needed if you did a sibling oocyte comparison, as suggested in point 2.
6. Title not appropriate ‘Timing of ICSI with respect to meiotic spindle status” would be better. Unless you are looking at timing of 2PN appearance etc.
7. Oocyte utilisation not clear—is it all injected oocytes (or fertilised oocytes) which were either transferred or frozen (does everything get frozen on day of transfer? Pregnancy rate
8. Pregnancy detection: week 5, embryonic sac—isn’t it more appropriate to do week 6 -and presence of a foetal heart?
Author Response
RESPONSE TO REVIEWERS
We thank all reviewers very much for evaluating our manuscript and providing us with useful notes about how to improve the article. We completely agree with all the recommendations and have therefore modified the manuscript accordingly. Step-by-step responses are provided below. All the relevant changes are highlighted in yellow.
- Only those consenting to MS got MS. Those not consenting were controls. This would create bias. All those consenting should have been randomised between having MS and not having MS
In the group of patients in which the MS was determined by polarized microscopy the position of PB and MS together with MS visibility were used as indicators of oocyte maturity. It should be noted that patients with MS evaluated by polarized microscopy were randomly selected, and only the patients who signed the agreement (more than 90% of patients) with the spindle status observation formed the MS evaluated group. The control group was formed by the standard patients undergoing ICSI.
- If ‘measuring maturity’was an end point, then each oocyte should have been treated individually. Ie all oocytes with ‘the PB/MS in close proximity or the MS was not visible’ should have been split into early and late ICSIs, and paired within cohorts, and equally all oocytes with the ‘MS clearly visible and the PB and MS not in close proximity’ should have been split into the 2 ICSI times rather than the whole cohort (ie at least 40% oocytes with MS clearly visible, PB/MS not in close proximity).
Patients having more than 60% oocytes with PB/MS in close proximity (angle between PB and MS < 5°) (Fig. 3A; Fig. S2 and S3) or MS not visible (Fig. 3B; Fig S1) were predicted as immature oocytes. For these patients ICSI was performed 4-5 hours after the polarization microscopy evaluation, i.e. 7-8 hours after OPU. 2. Patients having at least 40% oocytes with MS clearly visible, PB/MS not in close proximity (angle between PB and MS > 5°) (Fig. 1C) were predicted as mature oocytes (Fig. 3C; Figs. S2 and S3). For these patients ICSI was performed typically 2-3 hours post the polarization microscopy evaluation i.e. 5-6 hours after OPU, as patients from control group. The reasons for this process strategy are operational: in our center it is required that all oocytes of one patient be fertilized at the same time/in one group.
- All oocytes should have been followed up with fert data (1PN, 2PN, 3PN, 0PN, syngamy)-this would give a much clearer picture of maturity and timing of actual fertilisation
Correct, but it should be emphasized that the main focus of our study was to evaluate the efficiency of using meiotic spindle visibility and relative position to the polar body, as indicators of oocyte maturation, to optimize intracytoplasmic sperm injection timing with respect to patient’s age. In our study the incidence of 3PN and 1PN was low (for 3PN 0,5-8,1%).
- The stats were not clear. I have never seen P<0.1 being classed as significant,
Even though the statistical relevance (p) was < 0.1 in the case of the differences in pregnancy rates between MS evaluated patients ≥ 35 with ICSI performed 5-6 hours after OPU and MS evaluated patients ≥ 35 with ICSI performed 7-8 hours after OPU (number of patients in the individual groups), we regard the results as clinically significant. It is important to continue this research with a bigger cohort and with the help of artificial intelligence and neural network learning to achieve advanced PB/MS evaluation.
- Apart from age and sperm quality, there were no other demographic comparisons of the patient groups (stim type, # previous cycles, type of infertility, # oocytes, #GVs and M1s within a cohort, day of transfer etc) Note: this would not be needed if you did a sibling oocyte comparison, as suggested in point 2.
Corrected.
Table S1. Correlation matrixes for selected data sets of patients ≥ 35 years with calculated correlation coefficients.
Table S2. Correlation matrixes for selected data sets of patients < 35 years with calculated correlation coefficients.
- Title not appropriate ‘Timing of ICSI with respect to meiotic spindle status” would be better. Unless you are looking at timing of 2PN appearance etc.
Corrected.
- Oocyte utilisation not clear—is it all injected oocytes (or fertilised oocytes) which were either transferred or frozen (does everything get frozen on day of transfer? Pregnancy rate
For transfer and freezing we used good quality embryos. The probability of obtaining high-quality embryos (utilization rate) and clinical pregnancy after first OPU was compared between these two subgroups [30]. We freeze embryos at our clinic on day 5 and day 6 (in 98% of cases).
- 8. Pregnancy detection: week 5, embryonic sac—isn’t it more appropriate to do week 6 -and presence of a foetal heart?
A clinical pregnancy was confirmed by ultrasound visualization of the gestational sac or heartbeat. In our study the signs of clinical pregnancy were detected four weeks after embryo transfer.
Yours Sincerely,
Prof. Irena Kratochvilova, Ph.D. and A/Prof. Katerina Komrskova, Ph.D.

Reviewer 3 Report
Timing of oocyte fertilization by intracytoplasmic sperm injection using evaluation of meiotic spindle status
The aim of the study was to evaluate the efficiency of a method to assess the optimal time for oocyte fertilization by ICSI with respect to the patient’s age.
Introduction:
Line 50-51; ….with regard to the first PB location at the time of intracytoplasmic sperm injection (ICSI)….
Line 70: the authors evaluated oocyte meiotic spindle morphology at intracytoplasmic sperm injection (ICSI) as a predictor of blastocyst ploidy. This sentence need to be revised, <previous author [29] evaluated oocyte meiotic spindle morphology at intracytoplasmic sperm injection (ICSI) as a predictor of blastocyst ploidy.
Materials and methods
Line 246-249: All 246 patients involved in our study (undergoing ICSI) were divided into two age….. < the authors should refrain from the use if this word unless were is impossible, check the entire document < All 246 patients involved in the study……>
Line 257- 258: Using an optical microscope with a Nikon CEE GmbH polarizing filter, the research team effectively determined the angle between PB and MS together with MS visibility in oocytes which were collected from patients. The sentence need to be revised < Using an optical microscope with a Nikon CEE GmbH polarizing filter, to determine the angle between PB and MS together….>
Line 276-277: Postponing the ICSI time by 2-3 hours (7-8 hours after OPU) for the purpose of sufficient maturation [21] of oocytes, was based on the strategy proposed in [2,13,15,20,28]. The sentence require reversion < Postponing the ICSI time by 2-3 hours (7-8 hours after OPU) for the purpose of sufficient oocytes maturation [21], was based on the previous findings [2,13,15,20,28]
Line 284: (B) the oocyte with not visible MS. <(B) the oocyte with no visible MS > or <(B) the oocyte without visible MS >
Line 289: We compared the probability of obtaining high-quality embryos (utilization rate) a…..
Line 313: CO2 <CO2>
Line 316: We………
Conclusion
Line 359: In this study we proved that the visualization of MS was useful parameter for assessment…… < The present study proved that the visualization…..>
Discussion
Line 194: We compared utilization and pregnancy rates of patients younger than…..
Line 197-199: According to our results the age of…. <The age of ……>
Line 220: Our
Line 228: reached M2 often… <MII>
The manuscript require some minor revision before it can be published.
Author Response
RESPONSE TO REVIEWERS
We thank all reviewers very much for evaluating our manuscript and providing us with useful notes about how to improve the article. We completely agree with all the recommendations and have therefore modified the manuscript accordingly. Step-by-step responses are provided below. All the relevant changes are highlighted in yellow.
Introduction:
Line 50-51; ….with regard to the first PB location at the time of intracytoplasmic sperm injection (ICSI)….
Corrected: L. Rienzi et al. [2, 10, 13, 14] studied the relationship between MS / PB position and oocyte potential after ICSI.
Line 70: the authors evaluated oocyte meiotic spindle morphology at intracytoplasmic sperm injection (ICSI) as a predictor of blastocyst ploidy.
This sentence need to be revised, <previous author [29] evaluated oocyte meiotic spindle morphology at intracytoplasmic sperm injection (ICSI) as a predictor of blastocyst ploidy.
Corrected: In [29] the authors evaluated oocyte meiotic spindle morphology as a predictor of blastocyst ploidy.
Materials and methods
Line 246-249: All 246 patients involved in our study (undergoing ICSI) were divided into two age….. < the authors should refrain from the use if this word unless were is impossible, check the entire document < All 246 patients involved in the study……>
Corrected: All patients involved in the study (undergoing ICSI) were divided into two age dependent groups: 1. patients older than 35 years (after their 35th birthdays) and younger than 40 years, 2. patients younger than 35 years.
Line 257- 258: Using an optical microscope with a Nikon CEE GmbH polarizing filter, the research team effectively determined the angle between PB and MS together with MS visibility in oocytes which were collected from patients. The sentence need to be revised < Using an optical microscope with a Nikon CEE GmbH polarizing filter, to determine the angle between PB and MS together….>
Corrected.
Line 276-277: Postponing the ICSI time by 2-3 hours (7-8 hours after OPU) for the purpose of sufficient maturation [21] of oocytes, was based on the strategy proposed in [2,13,15,20,28]. The sentence require reversion < Postponing the ICSI time by 2-3 hours (7-8 hours after OPU) for the purpose of sufficient oocytes maturation [21], was based on the previous findings [2,13,15,20,28]
Corrected.
Line 284: (B) the oocyte with not visible MS. <(B) the oocyte with no visible MS > or <(B) the oocyte without visible MS >
Corrected.
Line 289: We compared the probability of obtaining high-quality embryos (utilization rate) a…..
Corrected.
Line 313: CO2 <CO2>
Corrected.
Line 316: We………
Corrected.
Conclusion
Line 359: In this study we proved that the visualization of MS was useful parameter for assessment…… < The present study proved that the visualization…..>
Corrected
Discussion
Line 194: We compared utilization and pregnancy rates of patients younger than…..
Corrected.
Line 197-199: According to our results the age of…. <The age of ……>
Corrected.
Line 220: Our
Corrected.
Line 228: reached M2 often… <MII>
Corrected.
Yours Sincerely,
Prof. Irena Kratochvilova, Ph.D. and A/Prof. Katerina Komrskova, Ph.D.

Round 2
Reviewer 2 Report
Some effort has been made to clarify this paper, and hope my comments make for a better follow up study